# Copeptin and stress-induced hyperglycemia in critically ill patients: A prospective study

Lilian Rodrigues Henrique[1], Daisy Crispim[2,3], Tarsila Vieceli[4], Ariell Freires Schaeffer[1], Priscila Bellaver[3,5], Cristiane Bauermann Leitão[1,2,3], Tatiana Helena Rech[3,5]*

1 School of Medicine, Universidade Federal do Rio Grande do Sul, Porto Alegre, Rio Grande do Sul, Brazil, 2 Endocrine Division, Hospital de Clínicas de Porto Alegre, Porto Alegre, Rio Grande do Sul, Brazil, 3 Post-Graduate Program in Medical Sciences: Endocrinology, Universidade Federal do Rio Grande do Sul, Porto Alegre, Rio Grande do Sul, Brazil, 4 Internal Medicine Division, Hospital de Clínicas de Porto Alegre, Porto Alegre, Rio Grande do Sul, Brazil, 5 Intensive Care Unit, Hospital de Clínicas de Porto Alegre, Porto Alegre, Rio Grande do Sul, Brazil

* threch@hcpa.edu.br

**Data Availability Statement:** All relevant data are within the paper and its Supporting information files.

## Abstract

### Objectives

Copeptin, an equimolar indicator of serum antidiuretic hormone levels, has been associated with higher mortality in critically ill patients and with the development of diabetes in the general population. The aim of the present study was to investigate the association of copeptin levels with glycemic parameters in critically ill patients and to compare the time-course of copeptin in survivors and non-survivors.

### Design

Prospective cohort study.

### Patients

From June to October 2019, critically ill patients were prospectively enrolled and followed for 90 days.

### Measurements

Plasma copeptin levels were determined at intensive care unit (ICU) admission (copeptin T1), 24 h (copeptin T2), and 48 h (copeptin T3) after study entry. Blood glucose and glycated hemoglobin levels were measured. ICU, in-hospital, and 90-day mortality, and length of stay in the ICU and hospital were evaluated.

### Results

104 patients were included. No significant correlation was detected between copeptin levels and blood glucose (r = -0.17, p = 0.09), HbA1c (r = 0.01, p = 0.9), glycemic gap (r = -0.16, p = 0.11), and stress hyperglycemia ratio (r = -0.14, p = 0.16). Copeptin T3 levels were significantly higher in survivors than in non-survivors at hospital discharge (561 [370–856] vs

**Funding:** This work was supported by Fundo de Incentivo à Pesquisa e Ensino (FIPE), Hospital de Clínicas de Porto Alegre (project number 2019-0304), Coordenação de Aperfeiçoamento de Pessoal de Nível Superior (CAPES), and Fundação de Amparo à Pesquisa do Estado do Rio Grande do Sul (Project number FAPERGS/CNPq 12/2014 - PRONEX). DC and CBL received scholarships from Conselho Nacional de Desenvolvimento Científico e Tecnológico (CNPq; PQ). The funders had no role in study design, data collection and analysis, decision to publish, or preparation of the manuscript.

**Competing interests:** The authors have declared that no competing interests exist.

300 [231–693] pg/mL, p = 0.015) and at 90 days (571 [380–884] vs 300 [232–698] pg/mL, p = 0.03).

## Conclusions

No significant correlations were found between copeptin levels and glycemic parameters, suggesting that copeptin is not a relevant factor in the induction of hyperglycemia during critical illness. Copeptin levels at ICU day 3 were higher in survivors than in non-survivors.

## Introduction

Facing acute stress, the human body responds with a hypercatabolic state that reduces and redirects energy consumption, delays anabolism, and activates the immune response [1]. The main metabolic response mechanism is the activation of the hypothalamic-pituitary-adrenal axis, and several studies have shown higher levels of stress hormones during an acute injury [2, 3]. However, stress hormone hypersecretion is associated with worse outcomes in critically ill patients [4].

Arginine-vasopressin (AVP), also known as antidiuretic hormone, is a hypothalamic hormone involved in stress response. It exerts a potentiating action on corticotropin-releasing hormone and, consequently, on adrenocorticotropic hormone (ACTH) release [5]. AVP is primarily activated by changes in plasma osmolarity or more than 10% reduction in blood volume, playing a central role in fluid balance and blood pressure [6]. As a matter of fact, AVP is a marker for the endogenous stress levels of a patient and low levels of AVP have been correlated with longer hemodynamic dysfunction [7].

Reliable measurements of AVP are difficult and currently not available. Instead, CTproAVP (copeptin) is a C-terminal part of the pre-pro-vasopressin precursor that is highly stable. Copeptin is released in equimolar amounts with AVP under various physiological and pathological conditions. Therefore, copeptin can be used as a surrogate for AVP secretion [6]. Jochberger et al. showed plasma concentrations of copeptin to be significantly higher in critically ill patients compared to healthy individuals [8]. Studies suggest that copeptin levels are related to higher mortality in critically ill patients [9]. In addition, an association between copeptin levels and the severity of critical illness is suggested, especially in patients with sepsis or hemorrhagic shock [10].

During critical illness, the decline in organ function is triggered by inflammatory mediators and by acute-phase changes in hormones [11]. These changes are characterized by elevated levels of AVP, sick euthyroid syndrome, and critical illness-related corticosteroid insufficiency [11]. Thyroid and adrenal function in critical illness have been extensively evaluated, including serial measurements of these hormones during the course of critical illness [12]. However, no studies of serial measurements of AVP in a general population of critically ill patients have been performed to date.

Acute stress induces insulin resistance and glucose overproduction by several mechanisms, including elevated levels of inflammatory cytokines, growth hormone, cortisol, glucagon, and catecholamines [13]. Stress-induced hyperglycemia is strongly associated with unfavorable outcomes in critically ill patients [13–15]. Interestingly, copeptin is associated with hyperglycemia in chronic diseases [16]. AVP induces hepatic glycogenolysis through V1a receptors [17] and stimulates insulin, glucagon, and ACTH secretion through V1b receptors [18]. Copeptin is an independent predictor of type 2 diabetes mellitus [19]. In line with this, a recent study

showed that low water intake correlated with high copeptin levels and was associated with an increased risk of diabetes mellitus, while increasing water intake reduced both copeptin and glucose levels [20]. Thus, AVP might have a role in the induction of hyperglycemia during critical illness.

Within this context, the aim of the present study was to investigate the association of copeptin levels with glycemic parameters in critically ill patients. The primary endpoint was the correlation between copeptin and blood glucose levels. The secondary endpoint was time-course differences in copeptin levels between survivors and non-survivors.

## Material and methods

### Study population

This is a prospective cohort study. The research protocol (project number 2019–0304) was approved by the ethics committee of the Hospital de Clínicas de Porto Alegre. The study protocol is in agreement with the Declaration of Helsinki for studies including human participants. All participants or their legal guardian signed an informed consent form. From June to October 2019, critically ill adult patients (age >18 years) admitted to the intensive care unit (ICU) of a university hospital in southern Brazil were prospectively evaluated within 24 h of ICU admission. Exclusion criteria were use of AVP for any reason, pregnancy, diabetic ketoacidosis, hyperglycemic hyperosmolar state, sickle cell anemia and other hemoglobinopathies, massive transfusion, pituitary disease, traumatic brain injury, neurosurgery, any intracranial pathology that may alter pituitary hormone secretion, and absence of central venous catheter or an arterial line for blood sampling. Death or discharge from ICU within 24 h of ICU admission was also considered exclusion criteria.

Plasma copeptin levels were measured in a time-course manner during ICU stay: at the study entry (copeptin T1, ICU day 1), 24 h after study entry (copeptin T2, ICU day 2), and 48 h after study entry (copeptin T3, ICU day 3). In addition, blood glucose and glycated hemoglobin (HbA1c) measurements were performed at study entry. Clinical and laboratory data were recorded for all patients via electronic medical records, including age, sex, comorbidities, reason for ICU admission, fluid balance (defined as the difference in mL between the input and output of fluids over a defined period of time) at 24, 48, and 72 h, presence of circulatory shock and need for vasoactive drugs, need for mechanical ventilation (MV), need for renal replacement therapy (RRT), prescription of corticosteroids and insulin, and nutritional therapy. The Simplified Acute Physiology Score 3 (SAPS 3) was used to grade disease severity, ranging theoretically from a minimum of 0 points to a maximum of 217 points, with higher scores indicating greater severity [21].

Patients were considered as having diabetes mellitus based on previous diagnosis or if they have HbA1c levels ≥6.5% at admission. Hyperglycemia was defined based on the threshold proposed by the American Diabetes Association [22] for in-hospital hyperglycemia as any blood glucose measurement > 140 mg/dL at ICU admission [23]. Insulin therapy was started at the discretion of the ICU team, to reach a target glycemic control ranging from 140 to 180 mg/dL, subcutaneous or intravenously [23, 24]. Insulin NPH was used as a long acting and regular insulin as a short acting insulin. Intravenous insulin protocol was started in hemodynamic unstable patients when high doses of insulin were required to reach the glycemic target. In order to separate the effects of a chronically altered metabolic state from those of acute stress hyperglycemia, the glycemic gap and stress hyperglycemia ratio (SHR) were evaluated. The glycemic gap was calculated as the difference between serum blood glucose on ICU admission and estimated mean blood glucose derived from HbA1c at admission (ICU admission serum blood glucose – estimated mean blood glucose), and SHR was calculated as the ratio between

the same parameters [25]. The level of HbA1c was used to estimate the mean blood glucose concentration using the following formula: (28.7 x HbA1c) − 46.7 mg/dL [25]. For the analysis of copeptin levels, patients were categorized at admission by presence of hyperglycemia (defined as blood glucose ≥ 140 mg/dL), by glycemic gap (< or ≥ 80 mg/dL), and by SHR (< or ≥ 1.1) [25].

The outcomes of interest were adjudicated by 2 researchers (L.R.H and A.F.S) and included the following: ICU mortality, in-hospital mortality, 90-day mortality, and length of stay (LOS) in the ICU and in the hospital.

## Biochemical measurements

Blood samples (10 mL) for plasma copeptin determination were centrifuged at 3000 rpm for 20 min at 8°C and stored at −80°C. For analysis, the samples were removed from the freezer and kept in room temperature to allow gradual thawing before measurements. Plasma copeptin levels were determined by enzyme-linked immunosorbent assay (ELISA) according to the manufacturer's instructions (Elabscience, Houston, TX, USA). This ELISA kit is based on the principle of competitive ELISA, with a sensitivity of 18.75 pg/mL and a detection range of 31.25–2000 pg/mL. The concentration of human copeptin was determined by comparing the optical density of the samples to a standard curve.

Blood samples for glucose determination were collected in tubes with sodium fluoride, centrifuged for 10 min at 3670 rpm, and analyzed by a hexokinase enzymatic method on the Roche COBAS c702 system (Roche Diagnostics, Mannheim, Germany). For HbA1c determination, samples were collected in EDTA tubes, homogenized and analyzed on the Bio-Rad Variant II Turbo system (Bio-Rad, Hercules, CA, USA), and processed by high-performance liquid chromatography. The concentration of glucose was expressed as mg/dL and of HbA1c as percentage.

## Statistical analysis

Sample size was calculated to provide at least 80% power at an alpha error of 5% to detect a correlation of at least 0.3 between copeptin and blood glucose levels (primary outcome). The sample size was calculated at 85 patients. Additionally, to detect a 20% difference in copeptin levels between survivors and non-survivors (secondary outcome), considering a power of 80% and an alpha error of 5%, a sample size of 44 individuals per group was required. WinPepi version 11.65 (Brixton Health, Jerusalem, Israel) was used for sample size calculation.

Data are presented as mean and SD, median and interquartile range (IQR), or number of cases and percentages. Comparisons between groups were performed using Student's *t* test, Mann-Whitney U test, or chi-square test, as appropriate. Correlations between variables were calculated using Spearman's test. To assess the relative risks (RR) of the variables and outcomes of interest, univariate linear regression or logistic regression models were constructed depending on the characteristics of the variables/outcomes. The plasma copeptin levels of survivors and non-survivors were compared at different time points by generalized estimating equations (GEE). Statistical analyses were performed using SPSS 21.0 (SPSS Inc., Chicago, IL, USA) and Stata 12.0 (StataCorp LLC, College Station, TX, USA). Statistical significance was set at $p < 0.05$.

## Results

### Study population characteristics

Of 345 eligible patients admitted to the ICU during the study period, 104 were included (Fig 1). The main characteristics of the study population are summarized in Table 1. Most patients

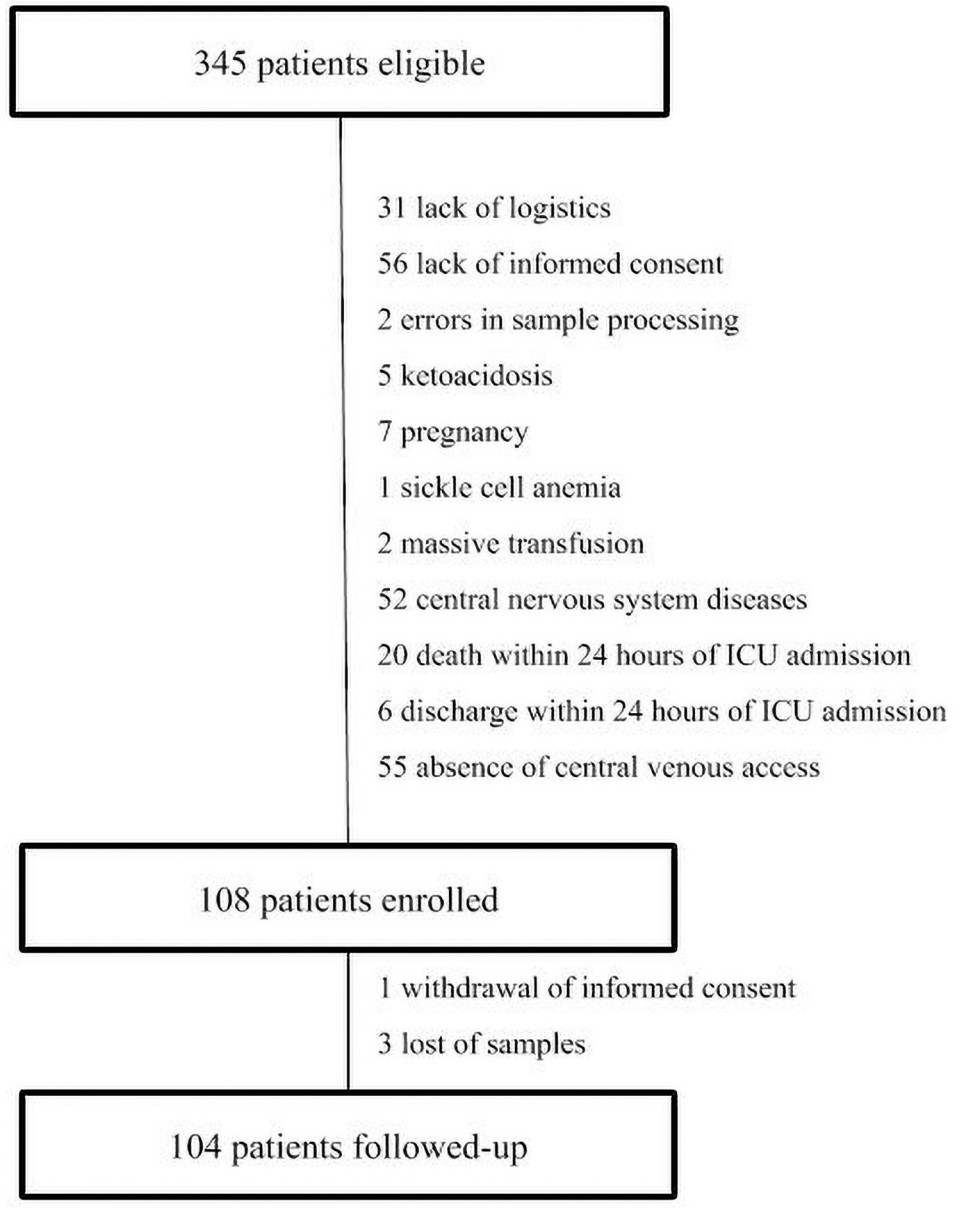

**Fig 1. Patient eligibility, enrollment, and follow-up.**

were men (54.8%), with a mean age of 62 (SD, 15) years and mean SAPS 3 score of 66 (SD, 17). The leading cause of ICU admission was hemodynamic shock (41.3%), followed by acute respiratory failure (31.7%) and postoperative care (16.3%). In addition, there was a high incidence of sepsis (68.3%). The most common comorbidities were hypertension (54.8%), diabetes mellitus (33.7%), and heart failure (32.7%). Briefly, 69.2% of patients (n = 72) required mechanical ventilation (MV) and 29.8% (n = 31) required renal replacement therapy (RRT). Overall ICU mortality was 26.9% (n = 28), in-hospital mortality was 37.5% (n = 39) and 90-day mortality was 40.4% (n = 42).

**Table 1. Baseline demographic and clinical characteristics at the time of intensive care unit admission.**

| Characteristics | Patients (n = 104) |
|---|---|
| **Demographics** | |
| Age (years) | 62 ± 15 |
| Male gender (n, %) | 57 (54.8) |
| **Comorbidities** | |
| Hypertension (n, %) | 57 (54.8) |
| Diabetes (n, %) | 35 (33.7) |
| Heart failure (n, %) | 34 (32.7) |
| Chronic kidney disease (n, %) | 23 (22.1) |
| COPD (n, %) | 11 (10.6) |
| Hypothyroidism (n, %) | 9 (8.7) |
| **Reasons for ICU admission** | |
| Shock (n, %) | 43 (41.3) |
| Acute respiratory failure (n, %) | 33 (31.7) |
| Postoperative care (n,%) | 17 (16.3) |
| Other (n, %) | 11 (10.6) |
| **Severity of Illness** | |
| SAPS 3 | 66 ± 17 |
| Presence of sepsis (n, %) | 71 (68.3) |
| Use of vasopressors (n, %) | 78 (75) |
| Need for MV (n, %) | 72 (69.2) |
| Use of corticosteroids (n, %) | 48 (46.2) |
| **Glycemic Parameters** | |
| Blood glucose (mg/dL) | 145 ± 53 |
| HbA1c (%) | 5.9 ± 1 |
| Glycemic gap (mg/dL) | 23 (12–45) |
| Stress hyperglycemia ratio | 1.22 ± 0.45 |
| Nutritional therapy (n, %) | 97 (92.3) |
| Insulin therapy (n, %) | 37 (35.6) |
| **Biochemical quantifications** | |
| Platelets ($10^3$/uL) | 182 (125–266) |
| Urea (mg/dL) | 69 (43–96) |
| Creatinine (mg/dL) | 1.6 (1.1–2.6) |
| Sodium (mEq/L) | 140 ± 5.5 |
| Potassium (mEq/L) | 4.4 ± 0.8 |
| Hematocrit (%) | 30.4 (24.7–36.3) |
| Hemoglobin (g/dL) | 10.2 ± 3.8 |
| Leukocytes ($10^3$/mm$^3$) | 10 (7.3–15.3) |
| PCR (mg/dL) | 93 (48–185) |
| Lactate (mmol/L) | 1.8 (1–3.4) |

BMI: body mass index; SAPS 3: simplified Acute Physiology III; COPD: chronic obstructive pulmonary disease; ICU: intensive care unit; HbA1c: glycated hemoglobin; MV: mechanical ventilation; PCR: C-reactive protein. Glycemic gap was calculated by the difference between the serum glucose at admission and the estimated mean blood glucose derived from HbA1c. Stress hyperglycemia ratio was defined by the ratio between serum glucose at admission and the estimated mean blood glucose derived from HbA1c. Values are mean ± standard deviation or median and interquartile range.

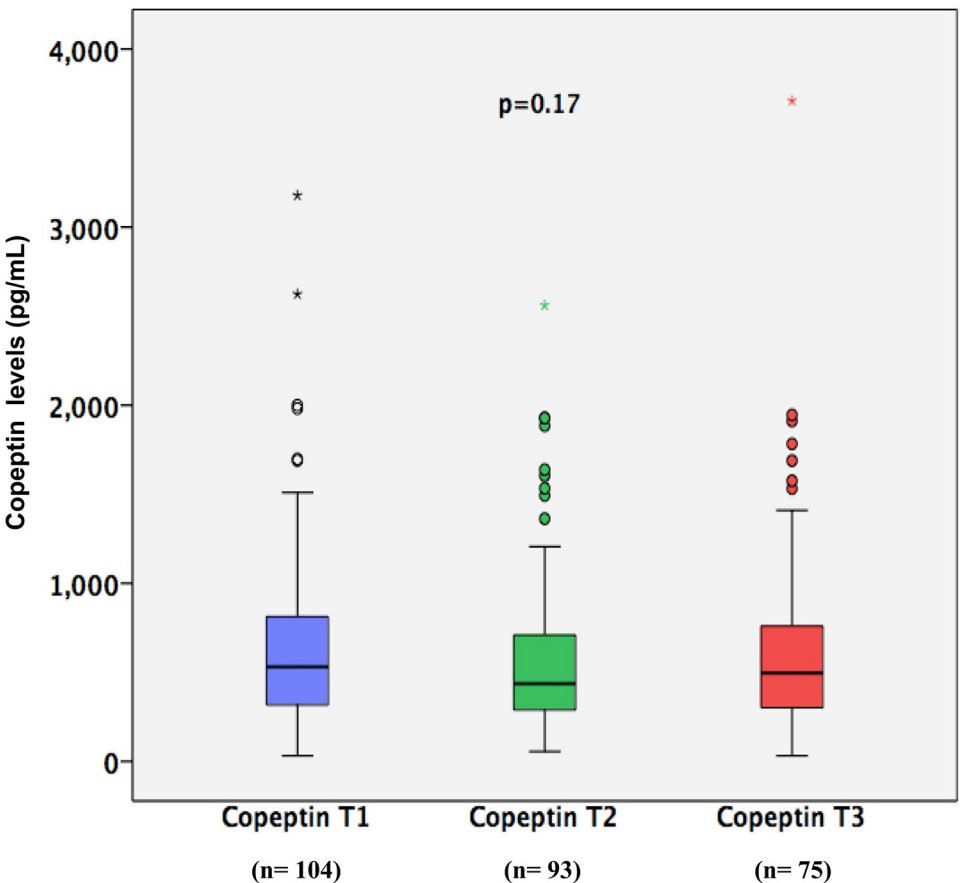

**Fig 2. Copeptin levels at three different time points.** T1: first time point of copeptin level (n = 104); T2: second time point of copeptin levels (n = 93); T3: third time point of copeptin levels (n = 75).

## Copeptin levels over time

Median copeptin levels did not differ between time points: 531 (IQR, 316–819) pg/mL at T1 (n = 104), 436 (IQR, 284–716) pg/mL at T2 (n = 93), and 496 (IQR, 301–792) pg/mL at T3 (n = 75) (p = 0.17) (Fig 2). Death (n = 9), discharge from ICU (n = 16), and sample hemolysis (n = 4) precluded the measurement of copeptin levels at all time points in 29 patients. Copeptin levels during the morning (6:00–12:00) were compared to copeptin levels during the day (12:00–6:00) at copeptin T1 340 (IQR, 259–574) vs 562 (IQR, 327–898) pg/mL, p = 0.02; at copeptin T2 399 (IQR, 329–503) vs 441 (IQR, 276–772) pg/mL, p = 0.39; and at copeptin T3 509 (IQR, 278–636) vs 491 (IQR, 301–884) pg/mL, p = 0.53. Only in the collection after admission to the ICU (T1) the copeptin collected during the day was smaller than that collected at night. The other measures, collected with the longest time since the onset of the critical illness, did not identify a difference.

## Copeptin levels and glycemic parameters

Twenty-two patients (21.1%) received at least one dose of subcutaneous insulin before admission to the ICU, and 37 patients (35.6%) received insulin during ICU stay as part of their clinical care at the discretion of the attending physician (subcutaneous or intravenous insulin). Mean blood glucose level was 145 (SD, 53) mg/dL, median glycemic gap was 23 (IQR, 12–45)

**Table 2. Association of copeptin levels with glycemic parameters.**

| Glycemic parameters | | Copeptin T1 (n = 104) | p |
|---|---|---|---|
| **Hyperglycemia** | >140 | 448 (314–702) | 0.17 |
| | <140 | 572 (362–939) | |
| **Glycemic gap** | >80 | 426 (305–959) | 0.90 |
| | <80 | 531 (329–804) | |
| **SHR** | >1.1 | 420 (312–624) | **0.01** |
| | <1.1 | 662 (408–968) | |

T1: first time point of copeptin level. SHR: stress hyperglycemia ratio. Hyperglycemia and glycemic gap are presented in mg/dL. Copeptin levels are present in pg/mL. Mann-Whitney U test was used. P value was considered significant at P <0.05.

mg/dL, and mean SHR was 1.21 (SD, 0.45) (Table 1). No significant correlation was detected between admission copeptin levels and blood glucose (r = -0.17, p = 0.09), HbA1c (r = 0.01, p = 0.90), glycemic gap (r = -0.16, p = 0.11), and SHR (r = -0.14, p = 0.16).

Interestingly, no differences were detected in copeptin levels between patients with (n = 35) or without diabetes (n = 69) at any time points at copeptin T1 444 (IQR, 301–770) vs 562 (IQR, 328–867) pg/mL, p = 0.31; at copeptin T2 383 (IQR, 247–623) vs 484 (IQR, 334–772) pg/mL, p = 0.09; at copeptin T3 484 (IQR, 299–684) vs 560 (IQR, 305–846) pg/mL, p = 0.55. Also, copeptin levels at T1 were compared according to glycemic parameters and the results are presented in Table 2. A low SHR (<1.1) were associated with higher copeptin levels (Table 2), however the association was lost after adjustment to the ICU severity score SAPS 3 in logistic regression analysis (RR 0.99 [95%CI 0.99–1.00], p = 0.12).

## Copeptin levels and fluid balance

Mean fluid balance was -18 (IQR, -833 to 1097) mL at 24, -108 (IQR, -1089 to 532) mL at 48, and -654 (IQR, -1607 to 122) mL at 72 h. There was no correlation between fluid balance at 24, 48, and 72 h and copeptin T1, T2, or T3 levels. Also, no correlation was detected between serum sodium and copeptin levels at T1 r = 0.096 p = 0.3, at T2 r = 0.11 p = 0.2 and at T3 r = 0.19 p = 0.09.

Additionally, there was no significant difference in copeptin levels between patients with and without acute kidney injury at copeptin T1 664 (IQR, 364–915) vs 453 (IQR, 312–777) pg/mL, p = 0.28; at copeptin T2 423 (IQR, 267–844) vs 436 (IQR, 289–709) pg/mL, p = 0.97); or at copeptin T3 589 (IQR, 308–1212) vs 496 (IQR, 301–727) pg/mL, p = 0.58.

## Copeptin levels and mortality

As expected, SAPS 3 score was significantly lower in survivors than in non-survivors (61 ± 16 vs 75 ± 15, p<0.001). However, no correlation was found between copeptin levels and SAPS 3: T1 (r = −0.08; p = 0.40), T2 (r = 0.02; p = 0.86), and T3 (r = −0.05; p = 0.60).

Copeptin levels did not differ between all survivors and non-survivors at copeptin T1 708 (IQR, 562–854) vs 572 (IQR, 438–706) pg/mL, p = 0.49; or at copeptin T2 631 (IQR, 499–764) vs 514 (IQR, 380–649) pg/mL, p = 0.47; but they were higher in survivors at copeptin T3 571 (IQR, 380–884) vs 300 (IQR, 232–698) pg/mL, p = 0.03 (Fig 3). Copeptin T3 levels were significantly higher in survivors at hospital discharge 561 (IQR, 370–856) vs 300 (IQR, 231–693) pg/mL, p = 0.01; and at 90 days 571 (IQR, 380–884) vs 300 (IQR, 232–698) pg/mL, p = 0.03. The associations of copeptin T1, T2, and T3 levels with mortality are summarized in Table 3. We

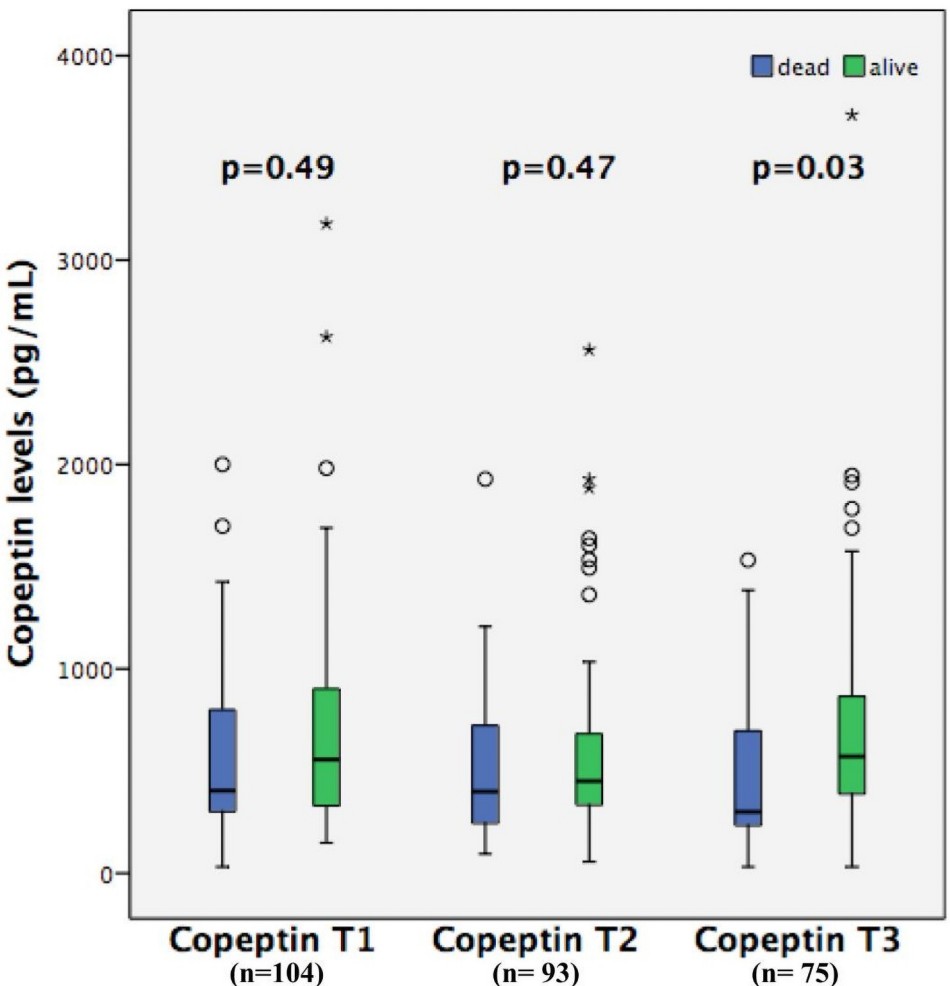

**Fig 3. Copeptin levels at three different time points in all survivors and non-survivors.** T1: first time point of copeptin level (n = 104); T2: second time point of copeptin levels (n = 93); T3: third time point of copeptin levels (n = 75).

performed a sensitivity analysis, looking at copeptin T1 and copeptin T2 values form the subset of patients who survived to collect copeptin T3 (n = 75). In this case, copeptin T1 and T2 showed similar results to copetin T3, being higher in survivals. Copeptin levels did not differ in patients with septic shock or in patients with shock from other causes at T1 386 [299–939] vs 421 [327–777] pg/mL, p = 0.73, at T2 467 [264–930] vs 429 [315–622] pg/mL, p = 0.81and at T3 506 [354–941] vs 491 [283–723] pg/mL, p = 0.57.

A receiver operating characteristic (ROC) curve analysis was performed to test the discriminatory power of copeptin to predict mortality. The area under the curve (AUC) was 0.55 (95% CI 0.43–0.67) at copeptin T1, 0.57 (95%CI 0.44–0.70) at copeptin T2, and 0.67 (95%CI 0.53–0.81) at copeptin T3. Considering the cut-off value of > 300 pg/mL for copeptin T3 (obtained from the ROC curve analysis), the sensitivity, specificity, positive predictive value, and negative predictive value were 50%, 90%, 72%, and 77%, respectively. Then, we divided patients into 2 groups according to copeptin T3 levels (≤ or >300 pg/mL) to test the association with mortality. At ICU day 3, patients with copeptin levels >300 pg/mL have a lower ICU mortality (50 vs 12.3%, p = 0.001), hospital mortality (72.2 vs 22.8%, p<0.001), and 90-day mortality (77.8 vs

**Table 3. Copeptin levels and mortality.**

| Outcomes | All (n = 104) | | Copeptin T1 | P | Copeptin T2 | P | Copeptin T3 | p |
|---|---|---|---|---|---|---|---|---|
| ICU mortality (n,%) | 28 (26.9) | Survivors (n = 59) | 534 (325 to 777) | 0.40 | 436 (329 to 655) | 0.9 | 517 (351 to 792) | 0.11 |
| | | Non-survivors (n = 16) | 378 (316 to 930) | | 422 (211 to 851) | | 297 (233 to 1103) | |
| Hospital mortality (n,%) | 39 (37.5) | Survivors (n = 49) | 622 (367 to 847) | 0.40 | 458 (367 to 727) | 0.27 | 561 (370 to 856) | **0.01** |
| | | Non-survivors (n = 26) | 363 (218 to 720) | | 396 (210 to 735) | | 300 (231 to 693) | |
| 90-day (n,%) | 42 (40.4) | Survivors (n = 46) | 623 (365 to 911) | 0.49 | 457 (348 to 752) | 0.47 | 571 (380 to 884) | **0.03** |
| | | Non-survivors (n = 28) | 367 (239 to 667) | | 398 (218 to 704) | | 300 (232 to 698) | |

T1: first time point of copeptin level; T2: second time point of copeptin levels; T3: third time point of copeptin levels. Mann-Whitney U test was used. P value was considered significant at P <0.05. Results are not adjusted for multiple comparisons. Copeptin values are present in pg/mL.

24.6, p<0.001) than patients with copeptin levels ≤300 pg/mL. In multivariate analysis, which included SAPS 3, age, and diabetes mellitus as covariates, the association remained significant with hospital mortality (RR 2.3 [95%CI 2.7–37.3], p = 0.001) and at 90-day mortality (RR 2.5 [95%CI 3.2–49.9], p<0.001). Patients with copeptin levels >300 pg/mL did not have longer LOS in ICU compared to patients with copeptin levels ≤300 pg/mL 12 (IQR, 7–22] vs 11 (IQR, 6–14), p = 0.22; as well as they did not differ regarding LOS in hospital 38 (IQR, 22–64) vs 54 (IQR, 14–61), p = 0.97.

## Discussion

In this sample of critically ill patients, including one-third with diabetes, we found no significant clinical correlation between copeptin levels and the following glycemic parameters: mean blood glucose, HbA1c, glycemic gap, and SHR. Interestingly, copeptin levels at ICU day 3 were increased in survivors to hospital discharge as well as in survivors at 90 days.

Although studies of outpatients have demonstrated increased copeptin levels in patients with diabetes and poor glycemic control [26] and in healthy individuals with higher glucose levels [27], the present study in ICU patients does not support these findings. In our study no difference was identified in the distribution of copeptin levels between groups with or without diabetes and with or without hyperglycemia. In addition, no correlations were detected between copeptin levels and glycemic parameters, including those parameters related to stress-induced hyperglycemia, as glycemic gap and SHR. Considering the effect of AVP on different receptors, it is possible to understand, at least in part, the non-sustained association between copeptin levels and glycemic parameters. The stimulation of V1a receptors results in a stimulus to glycogenolysis and hepatic glucose output [17]. In turn, by activating V1b receptors, AVP stimulates the secretion of insulin and glucagon, hormones with opposite effects on blood glucose [17]. Considering that acute stress can impact long-term metabolic changes [28], it is possible that copeptin at baseline might correlate with blood glucose later in the hospitalization.

Stress-induced hyperglycemia in critical illness results from the interplay between endocrine, autonomic and endothelial mechanisms with variable responses across individuals [6]. Therefore, considering that glycemic control is a consequence of multiple factors interacting in a complex way in a particular individual, it is difficult to hold a single factor responsible for the major role in its determination. In the present study, copeptin levels showed no significant correlation with blood glucose, glycemic gap, and SHR, which suggests that copeptin is not a relevant determinant in the induction of hyperglycemia during critical illness. There is no single clear-cut pathophysiological explanation for stress-induced hyperglycemia in ICU patients as many inflammatory and hormonal factors seem to contribute, but our results suggest copeptin is not a part of it.

In contrast to previous studies [9, 29–32], our results did not show an association between copeptin levels on ICU admission and mortality. However, we did find copeptin levels at ICU day 3 to be higher in survivors than in non-survivors, which suggests that a sustained increase in copeptin levels might be necessary to recover from acute stress, as well as from the higher catabolic demands of critical illness. Glucocorticoids levels are often increased in the acute phase of critical illness, whereas ACTH levels are suppressed (ACTH/cortisol dissociation) [33]. Assumed mechanisms involve decreased hepatic inactivation, decreased renal excretion, lower levels of cortisol binding globulin, increased levels of bile acids or direct stimulation of cortisol synthesis, and release by inflammatory cytokines [34]. Also, it has been demonstrated that ACTH/cortisol dissociation has a negative impact on prognosis [33–35]. As glucocorticoids suppress AVP secretion, this hypercortisolemic state during critical illness might be a reason for the lower copeptin levels at day 3 in non-survivors.

In contrast to previous studies, our results did not show an association between copeptin levels at ICU admission and mortality. One possible explanation for this difference is the severity of disease, since our sample included a heterogeneous population of very sick patients, as demonstrated by high SAPS 3 scores (Table 1). While our cohort had miscellaneous reasons for ICU admission, Koch et al. [9] included patients with or without sepsis, Ristagno et al. [30] restricted their population to survivors of out-of-hospital cardiac arrest, and Seligman et al. [31] studied patients with ventilator-associated pneumonia. This is reassured by the fact that copeptin at all time points (T1, T2 and T3) were higher in survivals when the subset of patients surveying until T3 were analyzed. However, having higher copeptin levels at ICU day 3 was associated with a reduced mortality, but not with reduced LOS in ICU or LOS in hospital. Besides, other important aspects of critical illness that may be related to copeptin remain unclear in the literature. Krychtiuk et al. [29] suggested that copeptin levels may be elevated in sepsis, but no association was observed with disease severity scores. On the other hand, Koch et al. [9] suggested that higher copeptin was associated with disease severity but not with sepsis. Although SAPS 3 score was higher in non-survivors, it did not correlate with copeptin levels. In sum, the heterogeneity of these results suggests that a comprehensive knowledge of the role of copeptin during critical illness is lacking, dictating the need for further evaluation.

This is the first prospective study to evaluate copeptin levels with glycemic parameters, but the study also has limitations. First, there were inevitable losses to follow-up at T2 (10%) and T3 (22%) due to death or discharge from the ICU, before patients had completed the time course of the study; however, the statistical test used to compare the curve time points considered this information. Second, volume status and fluid responsiveness were not assessed. The main role of AVP is to induce water conservation by the kidney in order to maintain osmotic and cardiovascular homeostasis [26]. The strongest stimulus to AVP release is, in fact, the changes in plasma osmolality, which were also not measured in the study. However, no correlation was found between copeptin levels and fluid balance, sodium levels, or acute kidney injury. Third, copeptin was measured at a varying time points during the day, which might have affected copeptin levels, since, as AVP, copeptin may have a small diurnal variation in healthy individuals [36]. However, critically ill patients frequently have abnormal circadian rhythms, as suggested by the lower levels of copeptin in the morning. Fourth, insulin levels were not measured, which might have been important to clarify a possible involvement of copeptin in insulin resistance. Fifth, glycemic parameters were measured once at the admission of patients to the ICU, making an interesting cross-sectional analysis impossible at a later time.

In summary, in this mixed medical-surgical series of 104 critically ill patients, including patients with and without diabetes, no significant correlations were found between copeptin levels and glycemic parameters, which suggest that copeptin is not a relevant determinant of

stress-induced hyperglycemia during critical illness. Of note, copeptin levels at ICU day 3 were significantly higher in survivors than in non-survivors, an interesting result that requires further investigation. Based on previous studies and on ours results, we believe that a more comprehensive, understanding of the role of copeptin during critical illness is far from being achieved.

## Supporting information

**S1 File.**
(XLSX)

## Acknowledgments

We thank Édison Moraes Rodrigues Filho for his assistance with SAPS 3 data acquisition.

## Author Contributions

**Conceptualization:** Lilian Rodrigues Henrique, Cristiane Bauermann Leitão, Tatiana Helena Rech.

**Data curation:** Lilian Rodrigues Henrique, Ariell Freires Schaeffer.

**Formal analysis:** Lilian Rodrigues Henrique, Daisy Crispim, Tarsila Vieceli, Ariell Freires Schaeffer, Priscila Bellaver, Cristiane Bauermann Leitão, Tatiana Helena Rech.

**Funding acquisition:** Daisy Crispim, Cristiane Bauermann Leitão, Tatiana Helena Rech.

**Investigation:** Lilian Rodrigues Henrique, Ariell Freires Schaeffer, Priscila Bellaver.

**Methodology:** Lilian Rodrigues Henrique, Daisy Crispim, Ariell Freires Schaeffer, Cristiane Bauermann Leitão, Tatiana Helena Rech.

**Project administration:** Tatiana Helena Rech.

**Software:** Tarsila Vieceli, Cristiane Bauermann Leitão.

**Supervision:** Daisy Crispim, Tatiana Helena Rech.

**Validation:** Tarsila Vieceli, Priscila Bellaver.

**Writing – original draft:** Lilian Rodrigues Henrique, Ariell Freires Schaeffer, Priscila Bellaver.

**Writing – review & editing:** Lilian Rodrigues Henrique, Daisy Crispim, Tarsila Vieceli, Priscila Bellaver, Cristiane Bauermann Leitão, Tatiana Helena Rech.

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
