## [Decision Letter · Decision Letter 0]

24 Feb 2021

PONE-D-20-40894

Copeptin and stress-induced hyperglycemia in critically ill patients: a prospective study

PLOS ONE

Dear Dr. Rech,

Thank you for submitting your manuscript to PLOS ONE. After careful consideration, we feel that it has merit but does not fully meet PLOS ONE’s publication criteria as it currently stands. Therefore, we invite you to submit a revised version of the manuscript that addresses the points raised during the review process.

We look forward to receiving your revised manuscript.

Kind regards,

Antonio Palazón-Bru, PhD

Academic Editor

PLOS ONE

Journal Requirements:

2) We note that you have included the phrase “data not shown” in your manuscript. Unfortunately, this does not meet our data sharing requirements. PLOS does not permit references to inaccessible data. We require that authors provide all relevant data within the paper, Supporting Information files, or in an acceptable, public repository. Please add a citation to support this phrase or upload the data that corresponds with these findings to a stable repository (such as Figshare or Dryad) and provide and URLs, DOIs, or accession numbers that may be used to access these data. Or, if the data are not a core part of the research being presented in your study, we ask that you remove the phrase that refers to these data.

3) We note that you have indicated that data from this study are available upon request. PLOS only allows data to be available upon request if there are legal or ethical restrictions on sharing data publicly. For information on unacceptable data access restrictions, please see http://journals.plos.org/plosone/s/data-availability#loc-unacceptable-data-access-restrictions.

Reviewers' comments:

Reviewer's Responses to Questions

**Comments to the Author**

1. Is the manuscript technically sound, and do the data support the conclusions?

Reviewer #1: Yes

Reviewer #2: Yes

Reviewer #3: Yes

2. Has the statistical analysis been performed appropriately and rigorously? 

Reviewer #1: Yes

Reviewer #2: Yes

Reviewer #3: Yes

3. Have the authors made all data underlying the findings in their manuscript fully available?

Reviewer #1: Yes

Reviewer #2: Yes

Reviewer #3: Yes

4. Is the manuscript presented in an intelligible fashion and written in standard English?

Reviewer #1: Yes

Reviewer #2: Yes

Reviewer #3: Yes

5. Review Comments to the Author

Reviewer #1: Thank you for the opportunity to review this manuscript submitted by Rech et al entitled “Copeptin and stress-induced hyperglycemia in critically ill patients: a prospective study.” This is a well-designed and well-written study examining the association between copeptin and hyperglycemia in critical illness, a pathway that has not previously been studied in this fashion in acute dysglycemia. Below are my recommendations for the submission.

Major Concerns:

1. Methods- The association between copeptin and blood glucose appears to have been measured at a single time point (T1). Would the authors have blood glucose measurements at the subsequent time points as well? This would be helpful for cross sectional analyses at the later time points.

2- Results- I found it difficult to interpret the findings associating copeptin at T3 with mortality given the lack of associations at T1 and T2 and the loss of follow up. The missing data is not at random. Would be helpful to have a sensitivity analysis where the last value is carried forward for patients missing data at T3 or where analyses at T1 and/or T2 are restricted to participants who survived to T3. These analyses would be informative either if positive or negative.

Minor Concerns:

1. Methods- Would request the authors provide information on the insulin correction protocols in the institution and comment whether patients may have received insulin prior to copeptin/glucose measurement as part of their clinical care.

2- Results- The incidence of acute kidney injury and use of renal replacement therapy appears to be high enough to influence results. Would the authors please comment on the potential for impairments in renal clearance to influence (or not influence) the copeptin results?

3- Results- Table 1- Organization of this table (eg- Demographics/Comorbidities/Severity of Illness/Lab Values/Glycemic Measures/etc) would be recommended

4- Results- Table 2- Please keep a consistent number of significant digits in the p values

5- Results- Would note on Table 3 that results are not adjusted for multiple comparisons.

6- Results- Please provide the results of the logistic regression analyses for SHR (unadjusted and adjusted, also what was this adjusted for?).

7- Discussion- The authors do a good job of explaining the findings in their study that do not find associations between copeptin and blood glucose, but I do not agree with the statement “Taken together, these findings suggest that copeptin and stress-induced hyperglycemia may be a result of the same underlying process, i.e. both variables are epiphenomena during a critical illness and not directly related to each other.” and I would recommend removing it. I would also consider the authors include a discussion that copeptin at baseline may be correlated with blood glucose later in the hospitalization but was not the focus of this current study.

Reviewer #2: I commend the authors on outstanding work with a solid hypothesis that copeptin levels would correlate with glycemic changes in critical illness. Although the findings were not as expected, it is essential that quality research like this is still published. I also applaud the authors for attempting to identify why copeptin levels may be correlated with mortality in critical illness. Using a patient sample rather than a genetically altered mouse is much preferred.

Reviewer #3: Copeptin and stress-induced hyperglycemia in critically ill patients: a prospective study

General: This study evaluated 104 patients and did not detect a correlation with copeptin levels and the following: blood glucose, glycemic gap, and stress hyperglycemia ratio. The levels of copeptin at 48 hours were significantly higher in patients surviving versus non-survivors at hospital discharge and 90 days.

Introduction

The introduction would benefit by discussion about Copeptin and AVP rather than hyperglycemia in the critically ill at the start. This allows the reader to better understand the connections.

References specially deal with sepsis and hemorrhagic shock for reference 12 and the introduction would benefit by stating this

Methods:

Why was central venous blood catheter required? Were there concerns with using a radial arterial line?

Was there time relation to the copeptin levels? That is, were they higher in the morning for instance?

This reviewer is unclear on why the outcomes would need to be adjudicated by 2 researchers.

Results

The leading cause of ICU admission was hemodynamic shock. In light of reference 12 were these hemorrhagic shock or septic shock cases? Was there any subanalysis for these potential causes?

Discussion

Again, the analysis of this data concerning causes of shock and whether that impacts results?

The discussion needs more explanation on why this study in contrast to previous studies did not show an association between ICU admission and mortality.

Why was this a higher factor?

6. PLOS authors have the option to publish the peer review history of their article (what does this mean?). If published, this will include your full peer review and any attached files.

Reviewer #1: No

Reviewer #2: No

Reviewer #3: No

---

## [Author Response · Author response to Decision Letter 0]

13 Mar 2021

Emily Chenette

Editor-in-Chief 

Plos One

Dear Dr. Chenette,

 Please find enclosed the revised version of the manuscript entitled “Copeptin and stress-induced hyperglycemia in critically ill patients: a prospective study”.

The comments and suggestions of the reviewers were very useful and all queries were addressed as recommended. Modifications in the revised manuscript version were made as a marked-up copy that highlights changes made to the original version.

We believe that this revised version of manuscript is more comprehensive and that its quality was improved due to the suggestions of the reviewers.

 We look forward to hearing from you. Please do not hesitate to contact us if you require any additional information.

Sincerely,

Tatiana Helena Rech

Hospital de Clínicas de Porto Alegre

Rua Ramiro Barcelos 2350, Zip code 90035-003

Porto Alegre, Rio Grande do Sul, Brazil

Phone: + 55 51 33598223 Fax: +55 51 33598630

E-mail: threch@hcpa.edu.br

ORCID ID: 0000-0002-2430-0118

 

Reviewer #1: 

Thank you for the opportunity to review this manuscript submitted by Rech et al entitled “Copeptin and stress-induced hyperglycemia in critically ill patients: a prospective study.” This is a well-designed and well-written study examining the association between copeptin and hyperglycemia in critical illness, a pathway that has not previously been studied in this fashion in acute dysglycemia. Below are my recommendations for the submission.

Major Concerns:

Comment 1: Methods - The association between copeptin and blood glucose appears to have been measured at a single time point (T1). Would the authors have blood glucose measurements at the subsequent time points as well? This would be helpful for cross sectional analyses at the later time points.

Answer 1: We regret we did not plan to measure blood glucose at all time points when we designed the study. We completely agree with the reviewer that it would be an interesting cross sectional analyses at the later time points, but we do not have more blood samples to perform biochemical quantifications. We included this point as a limitation, as follows (Discussion, page 17, paragraph 2):

“Fifth, glycemic parameters were measured once at the admission of patients to the ICU, making an interesting cross-sectional analysis impossible at a later time.”

Comment 2: Results - I found it difficult to interpret the findings associating copeptin at T3 with mortality given the lack of associations at T1 and T2 and the loss of follow up. The missing data is not at random. Would be helpful to have a sensitivity analysis where the last value is carried forward for patients missing data at T3 or where analyses at T1 and/or T2 are restricted to participants who survived to T3. These analyses would be informative either if positive or negative.

Answer 2: The missing data in time points T2 and T3 was a major concern of the study design. The plasma copeptin levels were compared at different time points by generalized estimating equations (GEE), a statistical test used to compare the curve time points that considered this information (missing data). Besides, we followed the reviewer suggestion and performed analyses at T1 and T2 restricted to participants who survived to T3 (n=75). The results are in agreement with the results at Copeptin T3 (Table 3 of the main manuscript) and are presented below: 

1- Sensitivity analysis at Copeptin T1 restricted to participants who survived to Copeptin T3 (n=75)

ICU mortality: 378 [316-930] vs 534 [325-777] pg/mL, p=0.74

Hospital mortality: 363 [218-720] vs 621 [367-847] pg/mL, p=0.01

90-day mortality: 366 [239-667] vs 623 [365-911] pg/mL, p=0.05

2- Sensitivity analysis at Copeptin T2 restricted to participants who survived to Copeptin T3 (n=75) 

ICU mortality: 422 [211-851] vs 436 [328-654] pg/mL, p=0.73

Hospital mortality: 396 [210-735] vs 458 [367-727] pg/mL, p=0.01

90-day mortality: 398 [218-703] vs 398 [224-684] pg/mL, p=0.05

 A sentence was added to the main manuscript in page 12, paragraph 3:

“We performed a sensitivity analysis, looking at copeptin T1 and copeptin T2 values form the subset of patients who survived to collect copeptin T3 (n=75). In this case, copeptin T1 and T2 showed similar results to copetin T3, being higher in survivals”

Minor Concerns:

Comment 3: Methods - Would request the authors provide information on the insulin correction protocols in the institution and comment whether patients may have received insulin prior to copeptin/glucose measurement as part of their clinical care.

Answer 3: We included the required information regarding insulin protocols on page 6, paragraph 1, as follows:

“Insulin therapy was started at the discretion of the ICU team, to reach a target glycemic control ranging from 140 to 180 mg/dL, subcutaneous or intravenously 23, 24. Insulin NPH was used as a long acting and regular insulin as a short acting insulin. Intravenous insulin protocol was started in hemodynamic unstable patients when high doses of insulin were required to reach the glycemic target.”

“Twenty-two patients (21.1%) received at least one dose of subcutaneous insulin before admission to the ICU, and 37 patients (35.6%) received insulin during ICU stay as part of their clinical care at the discretion of the attending physician (subcutaneous or intravenous insulin).” This information was added in Results, page 10, paragraph 2.

Comment 4: Results - The incidence of acute kidney injury and use of renal replacement therapy appears to be high enough to influence results. Would the authors please comment on the potential for impairments in renal clearance to influence (or not influence) the copeptin results?

Answer 4: This is a very interesting point. However, there are no association between copeptin values and acute kidney injury, as stated in the Results (page 11, paragraph 3). 

“Additionally, there was no significant difference in copeptin levels between patients with and without acute kidney injury at T1 (664 [364-915] vs 453 [312-777] pg/mL, p=0.28), at T2 (423 [267-844] vs 436 [289-709] pg/mL, p=0.97), or at T3 (589 [308-1212] vs 496 [301-727] pg/mL, p=0.58).”

Moreover, following the reviewer comment, we tested the association between copeptin levels in survivors and non-survivors and the need for renal replacement therapy (RRT) at the three copeptin time points, and the results are summarized bellow:

Please refer to Table in "Response to Reviewers" to check this results.

Comment 5: Results - Table 1- Organization of this table (eg- Demographics/Comorbidities/Severity of Illness/Lab Values/Glycemic Measures/etc) would be recommended.

Answer 5: We reorganized Table 1 in sections as suggested by the reviewer. Please refer to Table in page 9.

Comment 6: Results- Table 2- Please keep a consistent number of significant digits in the p values.

Answer 6: The corrections were made accordingly on Table 2 and throughout the manuscript (please refer to Table 2).

Comment 7: Results - Would note on Table 3 that results are not adjusted for multiple comparisons.

Answer 7: Thanks for this suggestion. We included this information on the bottom of Table 3.

Comment 8: Results - Please provide the results of the logistic regression analyses for SHR (unadjusted and adjusted, also what was this adjusted for?).

Answer 8: The results of the logistic regression analyses for SHR were adjusted for SAPS 3 and are now presented on Results, page 11, paragraph 1. 

Comment 9: Discussion - The authors do a good job of explaining the findings in their study that do not find associations between copeptin and blood glucose, but I do not agree with the statement “Taken together, these findings suggest that copeptin and stress-induced hyperglycemia may be a result of the same underlying process, i.e. both variables are epiphenomena during a critical illness and not directly related to each other.” and I would recommend removing it. I would also consider the authors include a discussion that copeptin at baseline may be correlated with blood glucose later in the hospitalization but was not the focus of this current study.

Answer 9: We removed the sentence, as suggested by the reviewer. We agree and include this consideration on page 15, paragraph 1, as follows. A new reference was added to this statement as well (Loss SH, Nunes DSL, Franzosi OS, Salazar GS, Teixeira C, Vieira SRR. Chronic critical illness: are we saving patients or creating victims? Rev Bras Ter Intensiva. 2017.)

“Considering that acute stress can impact long-term metabolic changes 28, it is possible that copeptin at baseline might correlate with blood glucose later in the hospitalization.”

Reviewer #2: 

Comment 1: I commend the authors on outstanding work with a solid hypothesis that copeptin levels would correlate with glycemic changes in critical illness. Although the findings were not as expected, it is essential that quality research like this is still published. I also applaud the authors for attempting to identify why copeptin levels may be correlated with mortality in critical illness. Using a patient sample rather than a genetically altered mouse is much preferred.

Answer 1: We thank the reviewer for this nice comment to our dedicated work.

Reviewer #3: 

Copeptin and stress-induced hyperglycemia in critically ill patients: a prospective study

General: This study evaluated 104 patients and did not detect a correlation with copeptin levels and the following: blood glucose, glycemic gap, and stress hyperglycemia ratio. The levels of copeptin at 48 hours were significantly higher in patients surviving versus non-survivors at hospital discharge and 90 days.

Introduction

Comment 1: The introduction would benefit by discussion about Copeptin and AVP rather than hyperglycemia in the critically ill at the start. This allows the reader to better understand the connections.

References specially deal with sepsis and hemorrhagic shock for reference 12 and the introduction would benefit by stating this.

Answer 1: We agree with the reviewer. We amended the Introduction to start with copeptin and AVP rather than hyperglycemia, as suggested (please refer to Introduction page 3). 

Besides, a statement to clarify to which population of critically ill patients the sentence refers to was added to page 3, paragraph 3, as presented below:

 “In addition, an association between copeptin levels and the severity of critical illness is suggested, especially in patients with sepsis or hemorrhagic shock.”

Methods

Comment 2: 

Why was central venous blood catheter required? Were there concerns with using a radial arterial line?

Answer 2: Thanks the reviewer for this observation. This point was not clear stated. In fact, the exclusion criteria used was the absence of a central venous catheter or an arterial line for blood sampling. We amended accordingly in Methods, page 5, paragraph 1.

“Exclusion criteria were use of AVP for any reason, pregnancy, diabetic ketoacidosis, hyperglycemic hyperosmolar state, sickle cell anemia and other hemoglobinopathies, massive transfusion, pituitary disease, traumatic brain injury, neurosurgery, any intracranial pathology that may alter pituitary hormone secretion, and absence of central venous catheter or an arterial line for blood sampling.”

Comment 3: Was there time relation to the copeptin levels? That is, were they higher in the morning for instance? 

Answer 3: This is an interesting point. Copeptin may have a small diurnal variation in healthy individuals (Beglinger S. The Circadian Rhythm of Copeptin, the C-Terminal Portion of Arginine Vasopressin. J Biomark. 2017), but this behavior is unknown in critically ill patients, who frequently have abnormal circadian rhythms. In our study, copeptin was measured at a varying time points during the day. We performed a statistical analysis to test if copeptin levels were higher in the morning (6:00-12:00) than during the day (12:00-6:00), as presented bellow and included in Results on page 10 paragraph 1 and on Discussion page 17, paragraph 2:

Copeptin T1: 340 [259-574] (n=14) vs 562 [327-898] (n=91) pg/mL; p=0.02.

Copeptin T2: 399 [329-503] (n=12) vs 441 [276-772] pg/mL (n=81); p=0.39.

Copeptin T3: 509 [278-636] (n=9) vs 491 [301-884] pg/mL (n=66); p=0.53.

" Only in the collection after admission to the ICU (T1) the copeptin collected during the day was smaller than that collected at night. The other measures, collected with the longest time since the onset of the critical illness, did not identify a difference.”

“Third, copeptin was measured at a varying time points during the day, which might have affected copeptin levels, since, as AVP, copeptin may have a small diurnal variation in healthy individuals. However, critically ill patients frequently have abnormal circadian rhythms, as suggested by the lower levels of copeptin in the morning.”

Comment 4: This reviewer is unclear on why the outcomes would need to be adjudicated by 2 researchers.

Answer 4: The reviewer is correct. As the outcome was undoubtful (mortality), it did not need double check. This was an excess of scientific rigor. 

Results

Comment 5: The leading cause of ICU admission was hemodynamic shock. In light of reference 12 were these hemorrhagic shock or septic shock cases? Was there any subanalysis for these potential causes?

Answer 5: The study was conducted in a medical ICU that does not admit trauma patients. The main cause of shock in our population was sepsis. Of the 43 patients admitted due to shock, 26 had septic shock (63.4%), 8 had hypovolemic shock (19.5%) and 7 had cardiogenic shock (17.1%). No patient had hemorrhagic shock. However, following the reviewer suggestion, we compared copeptin levels in patients with septic shock with patients with shock from other causes. The results are presented bellow:

Copeptin T1 in patients with septic shock versus shock from other causes: 

386 [299-939] vs 421 [327-777] pg/mL, p=0.73

Copeptin T2 in patients with septic shock versus shock from other causes:

467 [264-930] vs 429 [315-622] pg/mL, p=0.81

Copeptin T3 in patients with septic shock versus shock from other causes:

506 [354-941] vs 491 [283-723] pg/mL, p=0.57

A sentence to state these results was added to Results page 12, paragraph 2, as follows:

“Copeptin levels did not differ in patients with septic shock or in patients with shock from other causes at T1 386 [299-939] vs 421 [327-777] pg/mL, p=0.73, at T2 467 [264-930] vs 429 [315-622] pg/mL, p=0.81and at T3 506 [354-941] vs 491 [283-723] pg/mL, p=0.57”

Discussion

Comment 6: Again, the analysis of this data concerning causes of shock and whether that impacts results?

Answer 6: Please refer to answer to comment 5 to check the statistical analyses according to shock subtype. Copeptin levels did not differ in patients with septic shock from patients with shock from other causes. Additionally, we compared copeptin levels in patients with and without sepsis. Copeptin levels at the three times did not differ between these groups, as presented below:

Copeptin T1: 643 [483-803] vs 705 [481-929] pg/mL, p = 0.60

Copeptin T2: 582 [453-710] vs 602 [414-789] pg/mL, p = 0.60 

Copeptin T3: 674 [497-852] vs 656 [456-856] pg/mL, p = 0.50

A sentence to state these results was added to Results page 12, paragraph 2, as follows:

“Copeptin levels did not differ in patients with septic shock or in patients with shock from other causes at T1 386 [299-939] vs 421 [327-777] pg/mL, p=0.73, at T2 467 [264-930] vs 429 [315-622] pg/mL, p=0.81and at T3 506 [354-941] vs 491 [283-723] pg/mL, p=0.57”

Comment 7: The discussion needs more explanation on why this study in contrast to previous studies did not show an association between ICU admission and mortality.

Why was this a higher factor?

Answer 7: This is an interesting point of our findings and should be better clarified in fact. We amended the discussion as follows (page 16, paragraph 2):

“In contrast to previous studies, our results did not show an association between copeptin levels at ICU admission and mortality. One possible explanation for this difference is the severity of disease, since our sample included a heterogeneous population of very sick patients, as demonstrated by high SAPS 3 scores (Table 1). While our cohort had miscellaneous reasons for ICU admission, Koch et al 9 included patients with or without sepsis, Ristagno et al 30 restricted their population to survivors of out-of-hospital cardiac arrest, and Seligman et al 31 studied patients with ventilator-associated pneumonia. This is reassured by the fact that copeptin at all time points (T1, T2 and T3) were higher in survivals when the subset of patients surveying until T3 were analyzed.”

---

## [Decision Letter · Decision Letter 1]

30 Mar 2021

Copeptin and stress-induced hyperglycemia in critically ill patients: a prospective study

PONE-D-20-40894R1

Dear Dr. Rech,

We’re pleased to inform you that your manuscript has been judged scientifically suitable for publication and will be formally accepted for publication once it meets all outstanding technical requirements.

Kind regards,

Antonio Palazón-Bru, PhD

Academic Editor

PLOS ONE

Additional Editor Comments (optional):

Reviewers' comments:

Reviewer's Responses to Questions

**Comments to the Author**

1. If the authors have adequately addressed your comments raised in a previous round of review and you feel that this manuscript is now acceptable for publication, you may indicate that here to bypass the “Comments to the Author” section, enter your conflict of interest statement in the “Confidential to Editor” section, and submit your "Accept" recommendation.

Reviewer #1: All comments have been addressed

Reviewer #2: All comments have been addressed

Reviewer #3: All comments have been addressed

2. Is the manuscript technically sound, and do the data support the conclusions?

Reviewer #1: Yes

Reviewer #2: Yes

Reviewer #3: Yes

3. Has the statistical analysis been performed appropriately and rigorously? 

Reviewer #1: Yes

Reviewer #2: Yes

Reviewer #3: I Don't Know

4. Have the authors made all data underlying the findings in their manuscript fully available?

Reviewer #1: Yes

Reviewer #2: Yes

Reviewer #3: Yes

5. Is the manuscript presented in an intelligible fashion and written in standard English?

Reviewer #1: Yes

Reviewer #2: Yes

Reviewer #3: Yes

6. Review Comments to the Author

Reviewer #1: I had the opportunity to review both the initial and revised version of this manuscript. I applaud the authors for their work on this revision and they have addressed all of my comments. I commend them on a very nicely executed clinical study.

Reviewer #2: All questions were answered from the other reviewers. I applaud the authors for their great work. Thank you.

Reviewer #3: The authors have addressed my comments. The introduction might benefit by additional description and not just rearranging the order however this is minor and would only make the manuscript stronger.

7. PLOS authors have the option to publish the peer review history of their article (what does this mean?). If published, this will include your full peer review and any attached files.

Reviewer #1: No

Reviewer #2: No

Reviewer #3: No

---

## [Editor Report · Acceptance letter]

12 Apr 2021

PONE-D-20-40894R1 

Copeptin and stress-induced hyperglycemia in critically ill patients:a prospective study 

Dear Dr. Rech:

I'm pleased to inform you that your manuscript has been deemed suitable for publication in PLOS ONE. Congratulations! Your manuscript is now with our production department. 

Kind regards, 

on behalf of

Dr. Antonio Palazón-Bru 

Academic Editor

PLOS ONE